# A state-space model of cross-region dynamic connectivity in MEG/EEG

**Ying Yang**[*]    **Elissa M. Aminoff**[†]    **Michael J. Tarr**[*]    **Robert E. Kass**[*]

[*]Carnegie Mellon University, [†]Fordham University

ying.yang.cnbc.cmu@gmail.com, {eaminoff@fordham, michaeltarr@cmu, kass@stat.cmu}.edu

## Abstract

Cross-region dynamic connectivity, which describes the spatio-temporal dependence of neural activity among multiple brain regions of interest (ROIs), can provide important information for understanding cognition. For estimating such connectivity, magnetoencephalography (MEG) and electroencephalography (EEG) are well-suited tools because of their millisecond temporal resolution. However, localizing source activity in the brain requires solving an under-determined linear problem. In typical two-step approaches, researchers first solve the linear problem with generic priors assuming independence across ROIs, and secondly quantify cross-region connectivity. In this work, we propose a one-step state-space model to improve estimation of dynamic connectivity. The model treats the mean activity in individual ROIs as the state variable and describes non-stationary dynamic dependence across ROIs using time-varying auto-regression. Compared with a two-step method, which first obtains the commonly used minimum-norm estimates of source activity, and then fits the auto-regressive model, our state-space model yielded smaller estimation errors on simulated data where the model assumptions held. When applied on empirical MEG data from one participant in a scene-processing experiment, our state-space model also demonstrated intriguing preliminary results, indicating leading and lagged linear dependence between the early visual cortex and a higher-level scene-sensitive region, which could reflect feedforward and feedback information flow within the visual cortex during scene processing.

## 1   Introduction

Cortical regions in the brain are anatomically connected, and the joint neural activity in connected regions are believed to underlie various perceptual and cognitive functions. Besides anatomical connectivity, researchers are particularly interested in the spatio-temporal statistical dependence across brain regions, which may vary quickly in different time stages of perceptual and cognitive processes. Descriptions of such spatio-temporal dependence, which we call *dynamic connectivity*, not only help to model the joint neural activity, but also provide insights to understand how information flows in the brain. To estimate dynamic connectivity in human brains, we need non-invasive techniques to record neural activity with high temporal resolution. Magnetoencephalography (MEG) and electroencephalography (EEG) are well-suited tools for such purposes, in that they measure changes of magnetic fields or scalp voltages, which are almost instantaneously induced by electric activity of neurons.

However, spatially localizing the source activity in MEG/EEG is challenging. Assuming the brain source space is covered by $m$ discrete points, each representing an electric current dipole generated by the activity of the local population of neurons, then the readings of $n$ MEG/EEG sensors can be approximated with a linear transformation of the $m$-dimensional source activity. The linear transformation, known as the *forward model*, is computed using Maxwell equations given the relative

positions of sensors with respect to the scalp (1). Typically $m \approx 10^3 \sim 10^4$ whereas $n \approx 10^2 \ll m$, so the *source localization problem* — estimating the source activity from the sensor data— is under-determined. Previous work has exploited various constraints or priors for regularization, including $L_2$ norm penalty (2; 3), sparsity-inducing penalty (4), and priors that encourage local spatial smoothness or temporal smoothness (5; 6; 7; 8).

When estimating dynamic connectivity from MEG/EEG recordings, especially among several pre-defined regions of interest (ROIs), researchers often use a two-step procedure: Step 1, estimating source activity using one of the common source localization methods, for example, the minimum norm estimate (MNE) that penalizes squared $L_2$ norm (2); Step 2, extracting the mean activity of source points within each ROI, and then quantifying the statistical dependence among the ROIs, using various methods ranging from pairwise correlations of time series to Granger causality and other extensions (9). However, most of the popular methods in Step 1 do not assume dependence across ROIs. For example, MNE assumes that all source points have independent and identical priors. Even in methods that assume auto-regressive structures of source activity (6; 8), only dependence on the one-step-back history of a source point itself and its adjacent neighbors is considered, while long-range dependence across ROIs is ignored. Biases due to these assumptions in Step 1 can not be adjusted in Step 2 and thus may result in additional errors in the connectivity analysis.

Alternatively, one can combine source localization and connectivity analysis jointly in one step. Two pioneering methods have explored this direction. The dynamical causal modeling (DCM (10)) assumes the source activity includes only one single current dipole in each ROI, and the ROI dipoles are modeled with a nonlinear, neurophysiology-informed dynamical system, where time-invariant coefficients describe how the current activity in each ROI is dependent on the history of all ROIs. Another method (11) does not use pre-defined ROIs, but builds a time-invariant multivariate auto-regressive (AR) model of all $m$ source points, where the AR coefficients are constrained by structural white-matter connectivity and sparsity-inducing priors. Both methods use static parameters to quantify connectivity, but complex perceptual or cognitive processes may involve fast changes of neural activity, and correspondingly require time-varying models of dynamic connectivity.

Here, we propose a new one-step state-space model, designed to estimate dynamic spatio-temporal dependence across $p$ given ROIs directly from MEG/EEG sensor data. We define the mean activity of the source points within each individual ROI as our $p$-dimensional state variable, and use a time-varying multivariate auto-regressive model to describe how much the activity in each ROI is predicted by the one-step-back activity in the $p$ ROIs. More specifically, we utilize the common multi-trial structure of MEG/EEG experiments, which gives independent observations at each time point and facilitates estimating the time-varying auto-regressive coefficients. Given the state variable at each time point, activities of source points within each ROI are modeled as independent Gaussian variables, with the ROI activity as the mean and a shared ROI-specific variance; activities of source points outside of all ROIs are also modeled as independent Gaussian variables with a zero mean and a shared variance. Finally, along with the forward model that projects source activity to the sensor space, we build a direct relationship between the state variables (ROI activities) and the sensor observations, yielding a tractable Kalman filter model. Comparing with the previous one-step methods (10; 11), the main novelty of our model is the time-varying description of connectivity. We note that the previous methods and our model all utilize specific assumptions to regularize the under-determined source localization problem. These assumptions may not always be satisfied universally. However, we expect our model to serve as a good option in the one-step model toolbox for researchers, when the assumptions are reasonably met. In this paper, we mainly compare our model with a two-step procedure using the commonly applied MNE method, on simulated data and in a real-world MEG experiment.

## 2 Model

**Model formulation**   In MEG/EEG experiments, researchers typically acquire multiple trials of the same condition and treat them as independent and identically distributed (i.i.d.) samples. Each trial includes a fixed time window of $(T+1)$ time points, aligned to the stimulus onset. Assuming there are $n$ sensors and $q$ trials, we use $\boldsymbol{y}_t^{(r)}$ to denote the $n$-dimensional sensor readings at time $t$ ($t = 0, 1, 2, \cdots, T$) in the $r$th trial ($r = 1, 2, \cdots, q$). To be more succinct, when alluding to the sensor readings in a generic trial without ambiguity, we drop the superscript $^{(r)}$ and use $\boldsymbol{y}_t$ instead; the same omission works for source activity and the latent ROI activity described below. We also

assume the mean of sensor data across trials is an $n \times (T + 1)$ zero matrix; this assumption can be easily met by subtracting the $n \times (T + 1)$ sample mean across trials from the data.

MEG and EEG are mainly sensitive to electric currents in the pyramidal cells, which are perpendicular to the folded cortical surfaces (12). Here we define the source space as a discrete mesh of $m$ source points distributed on the cortical surfaces, where each source point represents an electric current dipole along the local normal direction. If we use an $m$-dimensional vector $\boldsymbol{J}_t$ to denote the source activity at time $t$ in a trial, then the corresponding sensor data $\boldsymbol{y}_t$ has the following form

$$\textbf{sensor model (forward model):} \quad \boldsymbol{y}_t = \boldsymbol{G}\boldsymbol{J}_t + \boldsymbol{e}_t, \quad \boldsymbol{e}_t \overset{i.i.d}{\sim} \mathcal{N}(\boldsymbol{0}, \boldsymbol{Q}_e) \tag{1}$$

where the $n \times m$ matrix $\boldsymbol{G}$ describes the linear projection of the source activity into the sensor space, and the sensor noise, $\boldsymbol{e}_t$, is modeled as temporally independent draws from an $n$-dimensional Gaussian distribution $\mathcal{N}(\boldsymbol{0}, \boldsymbol{Q}_e)$. The noise covariance $\boldsymbol{Q}_e$ can be pre-measured using recordings in the empty room or in a baseline time window before experimental tasks.

Standard source localization methods aim to solve for $\boldsymbol{J}_t$ given $\boldsymbol{y}_t, \boldsymbol{G}$ and $\boldsymbol{Q}_e$. In contrast, our model aims to estimate dynamic connectivity among $p$ pre-defined regions of interest (ROIs) in the source space (see Figure 1 for an illustration). We assume at each time point in each trial, the current dipoles of the source points within each ROI share a common mean. Given $p$ ROIs, we have a $p$-dimensional state variable $\boldsymbol{u}_t$ at time $t$ in a trial, where each element represents the mean activity in one ROI. The state variable $\boldsymbol{u}_t$ follows a time-varying auto-regressive model of order 1

$$\textbf{ROI model:} \quad \begin{aligned} &\boldsymbol{u}_0 \sim \mathcal{N}(\boldsymbol{0}, \boldsymbol{Q}_0) \\ &\boldsymbol{u}_t = \boldsymbol{A}_t \boldsymbol{u}_{t-1} + \boldsymbol{\epsilon}_t, \quad \boldsymbol{\epsilon}_t \sim \mathcal{N}(\boldsymbol{0}, \boldsymbol{Q}), \quad \text{for } t = 1, \cdots, T. \end{aligned} \tag{2}$$

where $\boldsymbol{Q}_0$ is a $p \times p$ covariance matrix at $t = 0$, and $\boldsymbol{A}_t$s are the time-varying auto-regressive coefficients, which describe lagged dependence across ROIs. The $p$-dimensional Gaussian noise term $\boldsymbol{\epsilon}_t$ is independent of the past, with a zero mean and a covariance matrix $\boldsymbol{Q}$.

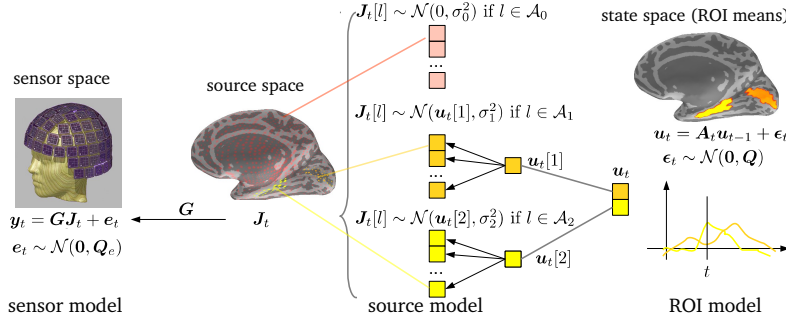

Figure 1: Illustration of the one-step state-space model

Now we describe how the source activity is distributed given the state variable (i.e., the ROI means). Below, we denote the $l$th element in a vector $\boldsymbol{a}$ by $\boldsymbol{a}[l]$, and the entry in $i$th row and $j$th column of a matrix $\boldsymbol{L}$ by $\boldsymbol{L}[i,j]$. Let $\mathcal{A}_i$ be the set of indices of source points in the $i$th ROI ($i = 1, 2, \cdots, p$); then for any $l \in \mathcal{A}_i$, the activity of the $l$th source point at time $t$ in a trial (scalar $\boldsymbol{J}_t[l]$) is modeled as the ROI mean plus noise,

$$\boldsymbol{J}_t[l] = \boldsymbol{u}_t[i] + \boldsymbol{w}_t[l], \quad \boldsymbol{w}_t[l] \overset{i.i.d.}{\sim} \mathcal{N}(0, \sigma_i^2), \quad \forall l \in \mathcal{A}_i, \tag{3}$$

where $\boldsymbol{w}_t$ denotes the $m$-dimensional noise on the $m$ source points given the ROI means $\boldsymbol{u}_t$, at time $t$ in the trial. Note that the mean $\boldsymbol{u}_t[i]$ is shared by all source points within the $i$th ROI, and the noise term $\boldsymbol{w}_t[l]$ given the mean is independent and identically distributed as $\mathcal{N}(0, \sigma_i^2)$ for all source points within the ROI, at any time in any trial. Additionally, we denote the indices of source points outside of any ROIs by $\mathcal{A}_0 = \{l, l \notin \cup_{i=1}^{p} \mathcal{A}_i\}$, and similarly, for each such source point, we also assume its activity at time $t$ in each trial has a Gaussian distribution, but with a zero mean, and a variance $\sigma_0^2$

$$\boldsymbol{J}_t[l] = 0 + \boldsymbol{w}_t[l], \quad \boldsymbol{w}_t[l] \overset{i.i.d.}{\sim} \mathcal{N}(0, \sigma_0^2), \quad \forall l \in \mathcal{A}_0. \tag{4}$$

We can concisely re-write (3) and (4) as

$$\textbf{source model:} \quad \boldsymbol{J}_t = \boldsymbol{L}\boldsymbol{u}_t + \boldsymbol{w}_t, \quad \boldsymbol{w}_t \overset{i.i.d.}{\sim} \mathcal{N}(\boldsymbol{0}, \boldsymbol{Q}_J) \tag{5}$$

where $\boldsymbol{L}$ is a $0/1$ $m \times p$ matrix indicating whether a source point is in an ROI (i.e., $\boldsymbol{L}[l, i] = 1$ if $l \in \mathcal{A}_i$ and $\boldsymbol{L}[l, i] = 0$ otherwise). The covariance $\boldsymbol{Q}_J$ is an $m \times m$ diagonal matrix, where each diagonal element is one among $\{\sigma_0^2, \sigma_1^2, \cdots \sigma_p^2\}$, depending on which region the corresponding source point is in; that is, $\boldsymbol{Q}_J[l, l] = \sigma_0^2$ if $l \in \mathcal{A}_0$ (outside of any ROIs), and $\boldsymbol{Q}_J[l, l] = \sigma_1^2$ if $l \in \mathcal{A}_1$, $\boldsymbol{Q}_J[l, l] = \sigma_2^2$ if $l \in \mathcal{A}_2$ and so on.

Combining the conditional distributions of $(\boldsymbol{y}_t | \boldsymbol{J}_t)$ given by (1) and $(\boldsymbol{J}_t | \boldsymbol{u}_t)$ given by (5), we can eliminate $\boldsymbol{J}_t$ (by integrating over all values of $\boldsymbol{J}_t$) and obtain the following conditional distribution for $(\boldsymbol{y}_t | \boldsymbol{u}_t)$

$$\boldsymbol{y}_t = \boldsymbol{C}\boldsymbol{u}_t + \boldsymbol{\eta}_t, \quad \boldsymbol{\eta}_t \overset{i.i.d.}{\sim} \mathcal{N}(\boldsymbol{0}, \boldsymbol{R}) \quad \text{where} \quad \boldsymbol{C} = \boldsymbol{G}\boldsymbol{L}, \quad \boldsymbol{R} = \boldsymbol{Q}_e + \boldsymbol{G}\boldsymbol{Q}_J\boldsymbol{G}' \quad (6)$$

where $\boldsymbol{G}'$ is the transpose of $\boldsymbol{G}$. Putting (2) and (6) together, we have a time-varying Kalman filter model, where the observed sensor data from $q$ trials $\{\boldsymbol{y}_t^{(r)}\}_{t=0,r=1}^{T,q}$ and parameters $\boldsymbol{Q}_e$, $\boldsymbol{G}$ and $\boldsymbol{L}$ are given, and the unknown set of parameters $\boldsymbol{\theta} = \{\{\boldsymbol{A}_t\}_{t=1}^T, \boldsymbol{Q}_0, \boldsymbol{Q}, \{\sigma_i^2\}_{i=0}^p\}$ are to be estimated. Among these parameters, we are mainly interested in $\{\boldsymbol{A}_t\}_{t=1}^T$, which describes the spatio-temporal dependence. Let $f(\cdot)$ denote probability density functions in general. We can add optional priors on $\boldsymbol{\theta}$ (denoted by $f(\boldsymbol{\theta})$) to regularize the parameters. For example, we can use $f(\boldsymbol{\theta}) = f(\{\boldsymbol{A}_t\}_{t=1}^T) \propto \exp(-(\lambda_0 \sum_{t=1}^T \|\boldsymbol{A}_t\|_F^2 + \lambda_1 \sum_{t=2}^T \|\boldsymbol{A}_t - \boldsymbol{A}_{t-1}\|_F^2))$, which penalizes the squared Frobenius norm ($\|\cdot\|_F$) of $\boldsymbol{A}_t$s and encourages temporal smoothness.

**Fitting the parameters using the expectation-maximization (EM) algorithm** To estimate $\boldsymbol{\theta}$, we maximize the objective function $\log f(\{\boldsymbol{y}_t^{(r)}\}_{t=0,r=1}^{T,q}; \boldsymbol{\theta}) + \log f(\boldsymbol{\theta})$ using the standard expectation-maximization (EM) algorithm (13). Here $\log f(\{\boldsymbol{y}_t^{(r)}\}_{t=0,r=1}^{T,q}; \boldsymbol{\theta})$ is the marginal log-likelihood of the sensor data, and $\log f(\boldsymbol{\theta})$ is the logarithm of the prior. We alternate between an E-step and an M-step.

In the E-step, given an estimate of the parameters (denoted by $\tilde{\boldsymbol{\theta}}$), we use the forward and backward steps in the Kalman smoothing algorithm (13) to obtain the posterior mean of $\boldsymbol{u}_t$, $\boldsymbol{u}_{t|T}^{(r)} \overset{\text{def}}{=} \mathbb{E}(\boldsymbol{u}_t^{(r)} | \{\boldsymbol{y}_\tau^{(r)}\}_{\tau=0}^T)$, the posterior covariance of $\boldsymbol{u}_t$, $\boldsymbol{P}_{t|T}^{(r)} \overset{\text{def}}{=} \text{cov}(\boldsymbol{u}_t^{(r)} | \{\boldsymbol{y}_\tau^{(r)}\}_{\tau=0}^T)$, and the posterior cross covariance of $\boldsymbol{u}_t$ and $\boldsymbol{u}_{t-1}$, $\boldsymbol{P}_{(t,t-1)|T}^{(r)} \overset{\text{def}}{=} \text{cov}(\boldsymbol{u}_t^{(r)}, \boldsymbol{u}_{t-1}^{(r)} | \{\boldsymbol{y}_\tau^{(r)}\}_{\tau=0}^T)$, for each $t$ in each trial $r$. Here $\mathbb{E}(\cdot)$ and $\text{cov}(\cdot)$ denote the expectation and the covariance. More details are in Appendix and (13).

In the M-step, we maximize the expectation of $\log f(\{\boldsymbol{y}_t^{(r)}\}_{t=0,r=1}^{T,q}, \{\boldsymbol{u}_t^{(r)}\}_{t=0,r=1}^{T,q}; \boldsymbol{\theta}) + \log f(\boldsymbol{\theta})$, with respect to the posterior distribution $\tilde{f} \overset{\text{def}}{=} f(\{\boldsymbol{u}_t^{(r)}\}_{t=0,r=1}^{T,q} | \{\boldsymbol{y}_t^{(r)}\}_{t=0,r=1}^{T,q}; \tilde{\boldsymbol{\theta}})$. Let $tr(\cdot)$ and $\det(\cdot)$ denote the trace and the determinant of a matrix. Given results in the E-step based on $\tilde{\boldsymbol{\theta}}$, the M-step is equivalent to minimizing three objectives separately

$$\min_{\boldsymbol{\theta}} (-\mathbb{E}_{\tilde{f}}(\log f(\{\boldsymbol{y}_t^{(r)}\}_{t=0,r=1}^{T,q}, \{\boldsymbol{u}_t^{(r)}\}_{t=0,r=1}^{T,q}; \boldsymbol{\theta})) - \log f(\boldsymbol{\theta}))$$
$$\equiv \min_{\boldsymbol{Q}_0} \mathcal{L}_1 + \min_{\boldsymbol{Q}, \{\boldsymbol{A}_t\}_{t=1}^T} \mathcal{L}_2 + \min_{\{\sigma_i^2\}_{i=0}^p} \mathcal{L}_3. \tag{7}$$

$$\mathcal{L}_1(\boldsymbol{Q}_0) = q \log \det(\boldsymbol{Q}_0) + tr(\boldsymbol{Q}_0^{-1} \boldsymbol{B}_0) \quad \text{where } \boldsymbol{B}_0 = \sum_{r=1}^q (\boldsymbol{P}_{0|T}^{(r)} + \boldsymbol{u}_{0|T}^{(r)} (\boldsymbol{u}_{0|T}^{(r)})') \tag{8}$$

$$\mathcal{L}_2(\boldsymbol{Q}, \{\boldsymbol{A}_t\}_{t=1}^T) = qT \log \det(\boldsymbol{Q}) + tr(\boldsymbol{Q}^{-1} \sum_{t=1}^T (\boldsymbol{B}_{1t} - \boldsymbol{A}_t \boldsymbol{B}_{2t}' - \boldsymbol{B}_{2t} \boldsymbol{A}_t' + \boldsymbol{A}_t \boldsymbol{B}_{3t} \boldsymbol{A}_t'))$$
$$+ \log f(\{\boldsymbol{A}_t\}_{t=1}^T) \tag{9}$$

$\quad$ where $\boldsymbol{B}_{1t} = \sum_{r=1}^q (\boldsymbol{P}_{t|T}^{(r)} + \boldsymbol{u}_{t|T}^{(r)} (\boldsymbol{u}_{t|T}^{(r)})'), \quad \boldsymbol{B}_{2t} = \sum_{r=1}^q (\boldsymbol{P}_{(t,t-1)|T}^{(r)} + \boldsymbol{u}_{t|T}^{(r)} (\boldsymbol{u}_{(t-1)|T}^{(r)})',)$

$\quad \boldsymbol{B}_{3t} = \sum_{r=1}^q (\boldsymbol{P}_{(t-1)|T}^{(r)} + \boldsymbol{u}_{(t-1)|T}^{(r)} (\boldsymbol{u}_{(t-1)|T}^{(r)})')$

$$\mathcal{L}_3(\{\sigma_i^2\}_{i=0}^p) = q(T+1) \log \det(\boldsymbol{R}) + tr(\boldsymbol{R}^{-1} \boldsymbol{B}_4), \quad \text{where } \boldsymbol{R} = \boldsymbol{Q}_e + \boldsymbol{G}\boldsymbol{Q}_J\boldsymbol{G}', \tag{10}$$

$\quad$ and $\boldsymbol{B}_4 = \sum_{r=1}^q \sum_{t=0}^T [(\boldsymbol{y}_t^{(r)} - \boldsymbol{C}\boldsymbol{u}_{t|T}^{(r)})(\boldsymbol{y}_t^{(r)} - \boldsymbol{C}\boldsymbol{u}_{t|T}^{(r)})' + \boldsymbol{C}\boldsymbol{P}_{t|T}^{(r)} \boldsymbol{C}']$

The optimization for the three separate objectives is relatively easy.

- For $\mathcal{L}_1$, the analytical solution is $\boldsymbol{Q}_0 \leftarrow (1/q)(\boldsymbol{B_0})$.

- For $\mathcal{L}_2$, optimization for $\{\boldsymbol{A}_t\}_{t=1}^T$ and $\boldsymbol{Q}$ can be done in alternations. Given $\{\boldsymbol{A}_t\}_{t=1}^T$, $\boldsymbol{Q}$ has the analytical solution $\boldsymbol{Q} \leftarrow 1/(qT)\sum_{t=1}^T(\boldsymbol{B}_{1t} - \boldsymbol{A}_t\boldsymbol{B}_{2t}' - \boldsymbol{B}_{2t}\boldsymbol{A}_t' + \boldsymbol{A}_t\boldsymbol{B}_{3t}\boldsymbol{A}_t')$. Given $\boldsymbol{Q}$, we use gradient descent with back-tracking line search (14) to solve for $\{\boldsymbol{A}_t\}_{t=1}^T$, where the gradients are $\frac{\partial \mathcal{L}_2}{\partial \boldsymbol{A}_t} = 2\boldsymbol{Q}^{-1}(-\boldsymbol{B}_{2t} + \boldsymbol{A}_t\boldsymbol{B}_{3t}) + 2\boldsymbol{D}_t$, $\boldsymbol{D}_t = \lambda_1(2\boldsymbol{A}_t - \boldsymbol{A}_{t+1} - \boldsymbol{A}_{t-1}) + \lambda_0\boldsymbol{A}_t$ for $t = 2, \cdots, T-1$, $\boldsymbol{D}_t = \lambda_1(\boldsymbol{A}_1 - \boldsymbol{A}_2) + \lambda_0\boldsymbol{A}_1$ for $t = 1$, and $\boldsymbol{D}_t = \lambda_1(\boldsymbol{A}_t - \boldsymbol{A}_{T-1}) + \lambda_0\boldsymbol{A}_t$ for $t = T$.

- For $\mathcal{L}_3$, we can also use gradient descent to solve for $\sigma_i$, with the gradient $\frac{\partial \mathcal{L}_3}{\partial \sigma_i} = tr((\frac{\partial \mathcal{L}_3}{\partial \boldsymbol{R}})'\frac{\partial \boldsymbol{R}}{\partial \sigma_i})$, where $\frac{\partial \mathcal{L}_3}{\partial \boldsymbol{R}} = \boldsymbol{R}^{-1} - \boldsymbol{R}^{-1}\boldsymbol{B}_4\boldsymbol{R}^{-1}$ and $\frac{\partial \boldsymbol{R}}{\partial \sigma_i} = 2\sigma_i\boldsymbol{G}[:, l \in \mathcal{A}_i]\boldsymbol{G}[:, l \in \mathcal{A}_i]'$. Here $\boldsymbol{G}[:, l \in \mathcal{A}_i]$ denotes the columns in $\boldsymbol{G}$ corresponding to source points in the $i$th region.

Because the E-M algorithm only guarantees to find a local optimum, we use multiple initializations, and select the solution that yields the best objective function $\log f(\{\boldsymbol{y}_t^{(r)}\}_{t=0,r=1}^{T,q}) + \log f(\boldsymbol{\theta})$ (see the appendix on computing $\log f(\{\boldsymbol{y}_t^{(r)}\}_{t=0,r=1}^{T,q}; \boldsymbol{\theta})$). The implementation of the model and the E-M algorithm in Python is available at `github.com/YingYang/MEEG_connectivity`.

**Visualizing the connectivity**  We visualize the lagged linear dependence between any pair of ROIs. According to the auto-regressive model in (2), given $\{\boldsymbol{A}_t\}_{t=1}^T$, we can characterize the linear dependence of ROI means at time $t + h$ on those at time $t$ by

$$\boldsymbol{u}_{t+h} = \tilde{\boldsymbol{A}}_{t,t+h}\boldsymbol{u}_t + \text{ noise independent of } \boldsymbol{u}_t$$

where $\tilde{\boldsymbol{A}}_{t,t+h} = \prod_{\tau=t+h}^{t+1}\boldsymbol{A}_\tau$, and in $\prod_{\tau=t+h}^{t+1}$, $\tau$ decreases from $t+h$ to $t+1$. For two ROIs indexed by $i_1$ and $i_2$, $\tilde{\boldsymbol{A}}_{t,t+h}[i_1, i_2]$ indicates the linear dependence of the activity in ROI $i_1$ at time $t + h$ on the activity in ROI $i_2$ at time $t$, where the linear dependence on the activity at time $t$ in other ROIs and ROI $i_1$ itself is accounted for; similarly, $\tilde{\boldsymbol{A}}_{t,t+h}[i_2, i_1]$ indicates the linear dependence of the activity in ROI $i_2$ at time $t + h$ on the activity in ROI $i_1$ at time $t$. Therefore, we can create a $T \times T$ matrix $\boldsymbol{\Delta}$ for any pair of ROIs ($i_1$ and $i_2$) to describe their linear dependence at any time lag: $\boldsymbol{\Delta}[t, t + h] = \tilde{\boldsymbol{A}}_{t,t+h}[i_2, i_1]$ ($i_1$ leading $i_2$) and $\boldsymbol{\Delta}[t + h, t] = \tilde{\boldsymbol{A}}_{t,t+h}[i_1, i_2]$ ($i_2$ leading $i_1$), for $t = 1, \cdots, T$ and $h = 1, \cdots, T - t - 1$.

## 3  Results

To examine whether our state-space model can improve dynamic connectivity estimation empirically, compared with the two-step procedure, we applied both approaches on simulated and real MEG data. We implemented the following two-step method as a baseline for comparison. In Step 1, we applied the minimum-norm estimate (MNE (2)), one of the most commonly used source localization methods, to estimate $\boldsymbol{J}_t$ for each time point in each trial. This is a Bayesian estimate assuming an $L_2$ prior on the source activity. Given $\boldsymbol{G}, \boldsymbol{Q}_e$ and a prior $\boldsymbol{J}_t \sim \mathcal{N}(\boldsymbol{0}, (1/\lambda)\boldsymbol{I}), \lambda > 0$ and the corresponding $\boldsymbol{y}_t$, the estimate is $\boldsymbol{J}_t \leftarrow \boldsymbol{G}'(\boldsymbol{G}\boldsymbol{G}' + \lambda\boldsymbol{Q}_e)^{-1}\boldsymbol{y}_t$. We averaged the MNE estimates for source points within each ROI, at each time point and in each trial respectively, and treated the averages as an estimate of the ROI means $\{\boldsymbol{u}_t\}_{t=0,r=1}^{T,q}$. In Step 2, according to the auto-regressive model in (2), we estimated $\boldsymbol{Q}_0, \{\boldsymbol{A}_t\}_{t=1}^T$ and $\boldsymbol{Q}$ by maximizing the sum of the log-likelihood and the logarithm of the prior ($\log f(\{\boldsymbol{u}_t\}_{t=0,r=1}^{T,q}) + \log f(\{\boldsymbol{A}_t\}_{t=1}^T)$); the maximization is very similar to the optimization for $\mathcal{L}_2$ in the M-step. Details are deferred to the appendix.

### 3.1  Simulation

We simulated MEG sensor data according to our model assumptions. The source space was defined as $m \approx 5000$ source points covering the cortical surfaces of a real brain, with 6.2 mm spacing on average, and $n = 306$ sensors were used. The sensor noise covariance matrix $\boldsymbol{Q}_e$ was estimated from real data. Two bilaterally merged ROIs were used: the pericalcarine area (ROI 1), and the parahippocampal gyri (ROI 2) (see Figure 2a). We selected these two regions, because they were of interest when we applied the models on the real MEG data (see Section 3.2). We generated the auto-regressive coefficients for $T = 20$ time points, where for each $\boldsymbol{A}_t$, the diagonal entries were

set to 0.5, and the off-diagonal entries were generated as a Morlet function multiplied by a random scalar drawn uniformly from the interval $(-1, 1)$ (see Figure 2b for an example). The covariances $\boldsymbol{Q}_0$ and $\boldsymbol{Q}$ were random positive definite matrices, whose diagonal entries were a constant $a$. The variances of source space noise $\{\sigma_i^2\}_{i=0}^p$ were randomly drawn from a Gamma distribution with the shape parameter being 2 and the scale parameter being 1. We used two different values, $a = 2$ and $a = 5$, respectively, where the relative strength of the ROI means compared with the source variance $\{\sigma_i^2\}_{i=0}^p$ were different. Each simulation had $q = 200$ trials, and 5 independent simulations for each $a$ value were generated. The unit of the source activity was nanoampere meter (nAm).

When running the two-step MNE method for each simulation, a wide range of penalization values ($\lambda$) were used. When fitting the state-space model, multiple initializations were used, including one of the two-step MNE estimates. In the prior of $\{\boldsymbol{A}_t\}_{t=1}^T$, we set $\lambda_0 = 0$ and $\lambda_1 = 0.1$. For the fitted parameters $\{\boldsymbol{A}_t\}_{t=1}^T$ and $\boldsymbol{Q}$ we defined the relative error as the Frobenius norm of the difference between the estimate and the true parameter, divided by the Frobenius norm of the true parameter (e.g., for the true $\boldsymbol{Q}$ and the estimate $\hat{\boldsymbol{Q}}$, the relative error was $\|\hat{\boldsymbol{Q}} - \boldsymbol{Q}\|_F/\|\boldsymbol{Q}\|_F$). For different two-step MNE estimates with different $\lambda$s, the smallest relative error was selected for comparison. Figure 2c and 2d show the relative errors and paired differences in errors between the two methods; in these simulations, the state-space model yielded smaller estimation errors than the two-step MNE method.

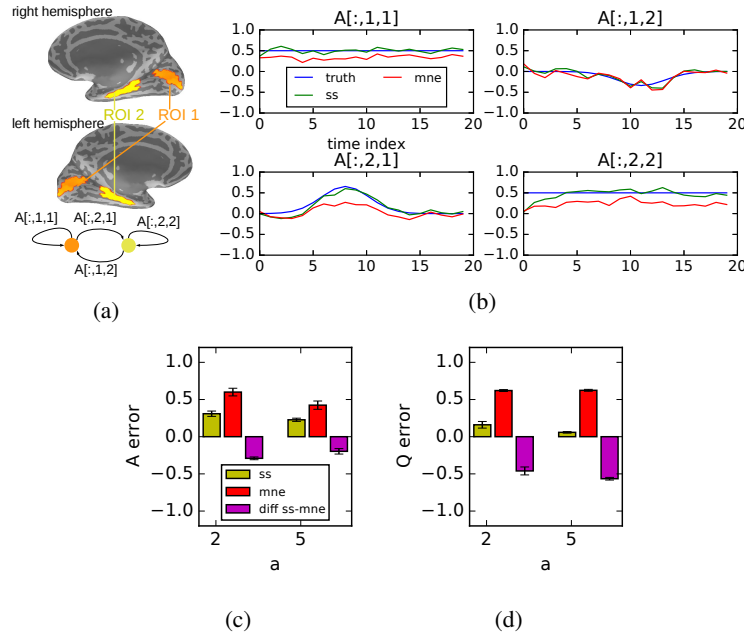

(a)         (b)

(c)         (d)

Figure 2: Simulation results. (a), Illustration of the two ROIs. (b), The auto-regressive coefficients $\{\boldsymbol{A}_t\}_{t=1}^T$ of $T = 20$ time points in one example simulation ($a = 5$). Here $\boldsymbol{A}[:, i_1, i_2]$ indicates the time-varying coefficient in $\boldsymbol{A}_t[i_1, i_2]$, for $i_1, i_2 = 1, 2$. (The legends: truth (blue), true values; ss (green), estimates by the state-space model; mne (red), estimates by the two-step MNE method.) (c) and (d), Comparison of the state-space model (ss) with the two-step MNE method (mne) in relative errors of $\{\boldsymbol{A}_t\}_{t=1}^T$ (c) and $\boldsymbol{Q}$ (d). The error bars show standard errors across individual simulations.

## 3.2 Real MEG data on scene processing

We also applied our state-space model and the two-step MNE method on real MEG data, to explore the dynamic connectivity in the visual cortex during scene processing. It is hypothesized that the ventral visual pathway, which underlies recognition of what we see, is organized in a hierarchical manner—along the pathway, regions at each level of the hierarchy receive inputs from previous levels, and perform transformations to extract features that are more and more related to semantics (e.g., categories of objects/scenes ) (15). Besides such feedfoward processing, a large number of top-down anatomical connections along the hypothesized hierarchy also suggest feedback effects

(16). Evidence for both directions has been reported previously (17; 18). However, details of the dynamic information flow during scene processing, such as when and how significant the feedback effect is, is not well understood. Here, as an exploratory step, we estimate dynamic connectivity between two regions in the ventral pathway: the early visual cortex (EVC) at the lowest level (in the pericalcarine areas), which is hypothesized to process low-level features such as local edges, and the parahippocampal place area (PPA), which is a scene-sensitive region on the higher level of the hierarchy and has been implicated in processing semantic information (19).

The 306-channel MEG data were recorded while a human participant was viewing 362 photos of various scenes. Each image was presented for 200 ms and repeated 5 times across the session, and data across the repetitions were averaged, resulting in $q = 362$ observations. The data was down-sampled from a sampling rate of 1 kHz to 100 Hz, and cropped within $-100 \sim 700$ Hz, where 0 ms marked the stimulus onset. Together, we had $T + 1 = 80$ time points (see the appendix for more preprocessing details). Given the data, we estimated the dynamic connectivity between the neural responses to the 362 images in the two ROIs (EVC and PPA), using our state-space model and the two-step MNE method. We created a source space including $m \approx 5000$ source points for the participant. In the prior of $\{A_t\}_{t=1}^{T}$, we set $\lambda_0 = 0$ and $\lambda_1 = 1.0$; in the two-step MNE method, we used the default value of the tuning parameter ($\lambda$) for single-trial data in the MNE-python software (20). After fitting $Q_0$, $\{A_t\}_{t=1}^{T}$ and $Q$, we computed the $\Delta$ matrix, as defined in Section 2, to visualize the lagged linear dependence between the two ROIs (EVC and PPA). We also bootstrapped the 362 observations 27 times to obtain standard deviations of entries in $\Delta$, and then computed a $z$-score for each entry, defined as the ratio between the estimated value and the bootstrapped standard deviation. Note that the sign of the source activity only indicates the direction of the electric current, so negative entries in $\Delta$ are as meaningful as positive ones. We ran two-tailed $z$-tests on the $z$-scores (assuming a standard normal null distribution); then we plotted the absolute values of the $z$-scores that passed a threshold where the $p$-value $< 0.05/(T^2)$, using the Bonferroni correction for $T^2$ comparisons in all the entries (Figure 3). Larger absolute values indicate more significant non-zero entries of $\Delta$, and more significant lagged linear dependence. As illustrated in Figure 3a, the lower right triangle of $\Delta$ indicates the linear dependence of PPA activity on previous EVC activity (EVC leading PPA, lower- to higher-level), whereas the upper left triangle indicates the linear dependence of EVC activity on previous PPA activity (PPA leading EVC, higher- to lower-level).

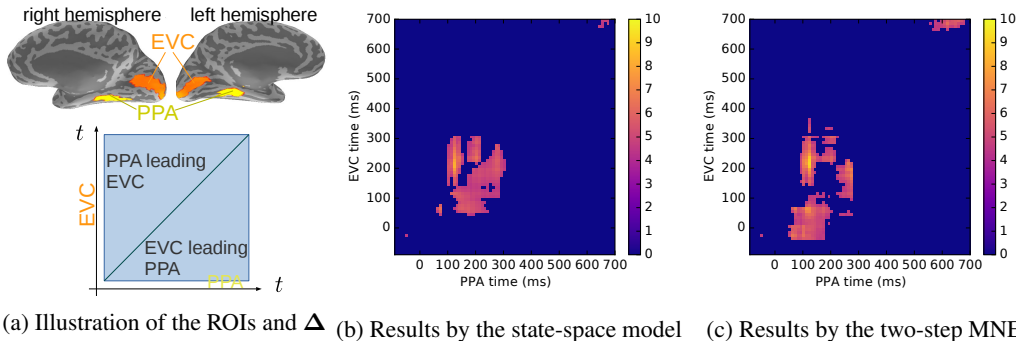

(a) Illustration of the ROIs and $\Delta$    (b) Results by the state-space model    (c) Results by the two-step MNE

Figure 3: Results from real MEG data on scene processing. (a), illustration of ROIs and the triangular parts of $\Delta$. (b) and (c), thresholded $z$-scores of $\Delta$ by the state-space model (b) and by the two-step MNE method (c).

Figure 3b and 3c show the thresholded absolute values of the $z$-scores by the state-space model and the two-step MNE method. In Figure 3b by the state-space model, we observed clusters indicating significant non-zero lagged dependence, in the lower right triangle, spanning roughly from 60 to 280 ms in EVC and from 120 to 300 ms in PPA, which suggests earlier responses in EVC can predict later responses in PPA in these windows. This pattern could result from feedforward information flow, which starts when EVC first receives the visual input near 60 ms. In the upper left triangle, we also observed clusters spanning from 100 to 220 ms in PPA and from 140 to 300 ms in EVC, suggesting earlier responses in PPA can predict later responses in EVC, which could reflect feedback along the top-down direction of the hierarchy. Figure 3c by the two-step MNE method also shows clusters in similar time windows, yet the earliest cluster in the lower right triangle appeared before 0 ms in EVC, which could be a false positive as visual input is unlikely to reach EVC that early.

We also observed a small cluster in the top right corner near the diagonal by both methods. This cluster could indicate late dependence between the two regions, but it was later than the typically evoked responses before 500 ms. These preliminary results were based on only one participant, and further analysis for more participants is needed. In addition, the apparent lagged dependence between the two regions are not necessarily direct or causal interactions; instead, it could be mediated by other intermediate or higher-level regions, as well as by the stimulus-driven effects. For example, the disappearance of the stimuli at 200 ms could cause an image-specific offset-response starting at 260 ms in the EVC, which could make it seem that image-specific responses in PPA near 120 ms predicted the responses at EVC after 260 ms. Therefore further analysis including more regions is needed, and the stimulus-driven effect needs to be considered as well. Nevertheless, the interesting patterns in Figure 3b suggest that our one-step state-space model can be a promising tool to explore the timing of feedforward and feedback processing in a data-driven manner, and such analysis can help to generate specific hypotheses about information flow for further experimental testing.

## 4    Discussion

We propose a state-space model to directly estimate the dynamic connectivity across regions of interest from MEG/EEG data, with the source localization step embedded. In this model, the mean activities in individual ROIs, (i.e., the state variable), are modeled with time-varying auto-regression, which can flexibly describe the spatio-temporal dependence of non-stationary neural activity. Compared with a two-step method, which first obtains the commonly used minimum-norm estimate of source activity, and then fits the auto-regressive model, our state-space model yielded smaller estimation errors than the two-step method in simulated data, where the assumptions in our model held. When applied on empirical MEG data from one participant in a scene-processing experiment, our state-space model also demonstrated intriguing preliminary results, indicating leading and lagged linear dependence between the early visual cortex and a higher-level scene-sensitive region, which could reflect feedforward and feedback information flow within the visual cortex. In sum, these results shed some light on how to better study dynamic connectivity using MEG/EEG and how to exploit the estimated connectivity to study information flow in cognition.

One limitation of the work here is that we did not compare with other one-step models (10; 11). In future work, we plan to do comprehensive empirical evaluations of the available one-step methods. Another issue is there can be violations of our model assumptions in practice. First, given the ROI means, the noise on source points could be spatially and temporally correlated, rather than independently distributed. Secondly, if we fail to include an important ROI, the connectivity estimates may be inaccurate—the estimates may not even be equivalent to the estimates when this ROI is marginalized out, due to the under-determined nature of source localization. Thirdly, the assumption that source points within an ROI share a common mean is typically correct for small ROIs but could be less accurate for larger ROIs, where the diverse activities of many source points might not be well-represented by a one-dimensional mean activity. That being said, as long as the activity in different source points within the ROI is not fully canceled, positive dependence effects of the kind identified by our model would still be meaningful in the sense that they reflect some cross-region dependence. To deal with the last two issues, one may divide the entire source space into sufficiently small, non-overlapping ROIs, when applying our state-space model. In such cases, the number of parameters can be large, and some sparsity-inducing regularization (such as the one in (11)) can be applied. In ongoing and future work, we plan to explore this idea and also address the effect of potential assumption violations.

**Acknowledgments**

This work was supported in part by the National Science Foundation Grant 1439237, the National Institute of Mental Health Grant RO1 MH64537, as well as the Henry L. Hillman Presidential Fellowship at Carnegie Mellon University.

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
