[Supplementary Material]

## Appendix

### Details of the E-M algorithm

Let $f(\cdot)$ denote probability density functions in general. We aim to obtain the set of parameters $\boldsymbol{\theta} = \{\{\boldsymbol{A}_t\}_{t=1}^T, \boldsymbol{Q}_0, \boldsymbol{Q}, \{\sigma_i^2\}_{i=0}^p\}$ that maximizes the objective function $\log f(\{\boldsymbol{y}_t^{(r)}\}_{t=0,r=1}^{T,q}; \boldsymbol{\theta}) + \log f(\boldsymbol{\theta})$, where $f(\boldsymbol{\theta})$ denotes the prior on $\boldsymbol{\theta}$. In our case $\log f(\{\boldsymbol{y}_t^{(r)}\}_{t=0,r=1}^{T,q}, \{\boldsymbol{u}_t^{(r)}\}_{t=0,r=1}^{T,q}; \boldsymbol{\theta})$ is much easier to compute than $\log f(\{\boldsymbol{y}_t^{(r)}\}_{t=0,r=1}^{T,q}; \boldsymbol{\theta})$, and the E-M algorithm (1) utilizes this property. Below, we briefly introduce how it works. Let $\tilde{\boldsymbol{\theta}}$ denote an estimate of $\boldsymbol{\theta}$. For a more succinct notation, let $\boldsymbol{u}^\dagger \stackrel{\text{def}}{=} \{\boldsymbol{u}_t^{(r)}\}_{t=0,r=1}^{T,q}$ and $\boldsymbol{y}^\dagger \stackrel{\text{def}}{=} \{\boldsymbol{y}_t^{(r)}\}_{t=0,r=1}^{T,q}$. Let $\tilde{f}(\boldsymbol{u}^\dagger) = f(\boldsymbol{u}^\dagger|\boldsymbol{y}^\dagger; \tilde{\boldsymbol{\theta}})$ be the posterior distribution of $\boldsymbol{u}^\dagger$ conditioned on observations $\boldsymbol{y}^\dagger$, based on the current estimate $\tilde{\boldsymbol{\theta}}$.

$$\log f(\boldsymbol{y}^\dagger; \boldsymbol{\theta}) + \log f(\boldsymbol{\theta})$$

$$= \int \tilde{f}(\boldsymbol{u}^\dagger) \log f(\boldsymbol{y}^\dagger; \boldsymbol{\theta}) d\boldsymbol{u}^\dagger + \log f(\boldsymbol{\theta}) = \mathbb{E}_{\tilde{f}}(\log f(\boldsymbol{y}^\dagger; \boldsymbol{\theta})) + \log f(\boldsymbol{\theta})$$

$$= \mathbb{E}_{\tilde{f}}(\log \frac{f(\boldsymbol{y}^\dagger, \boldsymbol{u}^\dagger; \boldsymbol{\theta})}{f(\boldsymbol{u}^\dagger|\boldsymbol{y}^\dagger; \boldsymbol{\theta})}) + \log f(\boldsymbol{\theta})$$

$$= \mathbb{E}_{\tilde{f}}(\log f(\boldsymbol{y}^\dagger, \boldsymbol{u}^\dagger; \boldsymbol{\theta})) + \log f(\boldsymbol{\theta}) - \mathbb{E}_{\tilde{f}}(\log f(\boldsymbol{u}^\dagger|\boldsymbol{y}^\dagger; \boldsymbol{\theta}))$$

This also holds for $\boldsymbol{\theta} = \tilde{\boldsymbol{\theta}}$.

$$\log f(\boldsymbol{y}^\dagger; \tilde{\boldsymbol{\theta}}) + \log f(\tilde{\boldsymbol{\theta}}) = \mathbb{E}_{\tilde{f}}(\log f(\boldsymbol{y}^\dagger, \boldsymbol{u}^\dagger; \tilde{\boldsymbol{\theta}})) + \log f(\tilde{\boldsymbol{\theta}}) - \mathbb{E}_{\tilde{f}}(\log \tilde{f}(\boldsymbol{u}^\dagger))$$

Now consider the difference

$$(\log f(\boldsymbol{y}^\dagger; \boldsymbol{\theta}) + \log f(\boldsymbol{\theta})) - (\log f(\boldsymbol{y}^\dagger; \tilde{\boldsymbol{\theta}}) + \log f(\tilde{\boldsymbol{\theta}})) \tag{1}$$

$$= \mathbb{E}_{\tilde{f}}(\log f(\boldsymbol{y}^\dagger, \boldsymbol{u}^\dagger; \boldsymbol{\theta})) + \log f(\boldsymbol{\theta}) \tag{2}$$

$$- \mathbb{E}_{\tilde{f}}(\log f(\boldsymbol{y}^\dagger, \boldsymbol{u}^\dagger; \tilde{\boldsymbol{\theta}})) - \log f(\tilde{\boldsymbol{\theta}}) \tag{3}$$

$$+ \mathbb{E}_{\tilde{f}}(\log \frac{\tilde{f}(\boldsymbol{u}^\dagger)}{f(\boldsymbol{u}^\dagger|\boldsymbol{y}^\dagger; \boldsymbol{\theta})}) \tag{4}$$

In each iteration, given $\tilde{\boldsymbol{\theta}}$, we select the new $\boldsymbol{\theta}$ that maximize the term (2). Because term (3) is fixed given $\tilde{\boldsymbol{\theta}}$, and term (4) is the non-negative Kullback-Leibler distance between $\tilde{f}(\boldsymbol{u}^\dagger)$ and $f(\boldsymbol{u}^\dagger|\boldsymbol{y}^\dagger; \boldsymbol{\theta})$, we are essentially maximizing a lower bound of the difference term (1). Therefore, when it converges, term (4) goes to zero, and we reach a local maximum of the objective function.

In each iteration, there are two steps: an E-step to compute $\tilde{f}$ or the expression for term (2) given the current $\tilde{\boldsymbol{\theta}}$, and an M-step to maximize the term (2) and update $\boldsymbol{\theta}$.

If we only have a prior on $\{\boldsymbol{A}_t\}_{t=1}^T$, (i.e., $f(\boldsymbol{\theta}) = f(\{\boldsymbol{A}_t\}_{t=1}^T) \propto \exp(-(\lambda_0 \sum_{t=1}^T \|\boldsymbol{A}_t\|_F^2 + \lambda_1 \sum_{t=2}^T \|\boldsymbol{A}_t - \boldsymbol{A}_{t-1}\|_F^2)))$, then term (3) has the following form up to some constant

$$\mathbb{E}_{\tilde{f}}(\log f(\boldsymbol{y}^\dagger, \boldsymbol{u}^\dagger; \boldsymbol{\theta})) + \log f(\boldsymbol{\theta}) \tag{5}$$

$$= -\frac{1}{2}(q \log \det(\boldsymbol{Q}_0) + \mathbb{E}_{\tilde{f}}(\sum_{r=1}^q \boldsymbol{u}_0^{(r)\prime} \boldsymbol{Q}_0^{-1} \boldsymbol{u}_0^{(r)}) \tag{6}$$

$$- \frac{1}{2}(qT \log \det(\boldsymbol{Q}) + \mathbb{E}_{\tilde{f}}(\sum_{t=1}^T \sum_{r=1}^q (\boldsymbol{u}_t^{(r)} - \boldsymbol{A}_t \boldsymbol{u}_{t-1}^{(r)})' \boldsymbol{Q}^{-1} (\boldsymbol{u}_t^{(r)} - \boldsymbol{A}_t \boldsymbol{u}_{t-1}^{(r)})) + \log f(\{\boldsymbol{A}_t\}_{t=1}^T) \tag{7}$$

$$- \frac{1}{2}(q(T+1) \log \det(\boldsymbol{R}) + \mathbb{E}_{\tilde{f}}(\sum_{t=0}^T \sum_{r=1}^q (\boldsymbol{y}_t^{(r)} - \boldsymbol{C}\boldsymbol{u}_t^{(r)})' \boldsymbol{R}^{-1} (\boldsymbol{y}_t^{(r)} - \boldsymbol{C}\boldsymbol{u}_t^{(r)})). \tag{8}$$

where $'$ denotes the transpose of a column vector or a matrix, and $\det(\cdot)$ denotes the determinant of a square matrix. To evaluate (6), (7), and (8), we need to compute the posterior mean and covariance of

$u_t$, as well as the cross covariance of $u_t$ and $u_{t-1}$ at each $t$ for each trial $(r = 1, 2, \cdots, q)$ given $\tilde{\theta}$:

$$u_{t|T}^{(r)} \stackrel{def}{=} \mathbb{E}(u_t^{(r)}|\{y_\tau^{(r)}\}_{\tau=0}^T),$$

$$P_{t|T}^{(r)} \stackrel{def}{=} \text{cov}(u_t^{(r)}|\{y_\tau^{(r)}\}_{\tau=0}^T),$$

$$P_{(t,t-1)|T}^{(r)} \stackrel{def}{=} \text{cov}(u_t^{(r)}, u_{t-1}^{(r)}|\{y_\tau^{(r)}\}_{\tau=0}^T)$$

To compute these values, we use the forward and backward steps in the Kalman smoothing algorithm (1). For simplicity, we drop the superscript $^{(r)}$ and the $\tilde{\ }$ symbol on $\tilde{\theta}$. We define the following terms

$$u_{t|s} \stackrel{def}{=} \mathbb{E}(u_t|y_0, y_1, \cdots, y_s),$$

$$P_{t|s} \stackrel{def}{=} \text{cov}(u_t|y_0.y_1, \cdots, y_s),$$

$$P_{(t,t-1)|s} \stackrel{def}{=} \text{cov}(u_t, u_{t-1}|y_0, y_1, \cdots, y_s).$$

In the forward step, we set $u_{0|0} = Q_0 C'(CQ_0 C' + R)^{-1} y_0$ and $P_{0|0} = Q_0 - Q_0 C'(CQ_0 C' + R)^{-1} CQ_0$. For $t = 1, 2, \cdots, T$, we have

$$u_{t|(t-1)} = A_t u_{(t-1)|(t-1)}$$

$$P_{t|(t-1)} = A_t P_{(t-1)|(t-1)} A_t' + Q$$

$$K_t \stackrel{def}{=} P_{t|(t-1)} C'(C P_{t|(t-1)} C' + R)^{-1}$$

$$u_{t|t} = u_{t|(t-1)} + K_t(y_t - Cu_{t|(t-1)})$$

$$P_{t|t} = P_{t|(t-1)} - K_t C P_{t|(t-1)}$$

In the backward step, for $t = T, T-1, \cdots, 1$

$$H_{t-1} \stackrel{def}{=} P_{(t-1)|(t-1)} A_t' P_{t|(t-1)}^{-1}$$

$$u_{(t-1)|T} = u_{(t-1)|(t-1)} + H_{t-1}(u_{t|T} - A_t u_{(t-1)|(t-1)})$$

$$P_{(t-1)|T} = P_{(t-1)|(t-1)} + H_{t-1}(P_{t|T} - P_{t|(t-1)})H_{t-1}'$$

and with $P_{(T,T-1)|T} = (I - K_T C)A_T P_{(T-1)|(T-1)}$, for $t = T-1, T-2, \cdots, 2$, we have

$$P_{(t-1,t-2)|T} = P_{(t-1)|(t-1)} H_{t-2}' + H_{t-1}(P_{(t,t-1)|T} - A_t P_{(t-1)|(t-1)})H_{t-2}'.$$

If we denote the terms (6), (7), and (8) with $\mathcal{L}_1$, $\mathcal{L}_2$ and $\mathcal{L}_3$ respectively, then using the posterior statistics above, we have

$$\mathcal{L}_1 = q \log \det(Q_0) + \mathbb{E}_{\tilde{f}}(\sum_{r=1}^q (u_0^{(r)'} Q_0^{-1} u_0^{(r)})) = q \log \det(Q_0) + tr(Q_0^{-1} B_0)$$

$$\mathcal{L}_2 = qT \log \det(Q) + \mathbb{E}_{\tilde{f}}(\sum_{t=1}^T \sum_{r=1}^q (u_t^{(r)} - A_t u_{t-1}^{(r)})' Q^{-1} (u_t^{(r)} - A_t u_{t-1}^{(r)}))] - \log f(\{A_t\}_{t=1}^T)$$

$$= qT \log \det(Q) + tr(Q^{-1} \sum_{t=1}^T (B_{1t} - A_t B_{2t}' - B_{2t} A_t' + A_t B_{3t} A_t')) + \lambda_0 \sum_{t=1}^T \|A_t\|_F^2 + \lambda_1 \sum_{t=2}^T \|A_t - A_{t-1}\|_F^2$$

$$\mathcal{L}_3 = q(T+1) \log \det(R) + \mathbb{E}_{\tilde{f}}(\sum_{t=0}^T \sum_{r=1}^q (y_t(r) - Cu_t^{(r)})' R^{-1} (y_t(r) - Cu_t^{(r)})')$$

$$= q(T+1) \log \det(R) + tr(R^{-1} B_4)$$

where

$$B_0 = \sum_{r=1}^{q}(P_{0|T}^{(r)} + u_{0|T}^{(r)}(u_{0|T}^{(r)})')$$

$$B_{1t} = \sum_{r=1}^{q}(P_{t|T}^{(r)} + u_{t|T}^{(r)}(u_{t|T}^{(r)})')$$

$$B_{2t} = \sum_{r=1}^{q}(P_{(t,t-1)|T}^{(r)} + u_{t|T}^{(r)}(u_{(t-1)|T}^{(r)})')$$

$$B_{3t} = \sum_{r=1}^{q}(P_{(t-1)|T}^{(r)} + u_{(t-1)|T}^{(r)}(u_{(t-1)|T}^{(r)})')$$

$$B_4 = \sum_{r=1}^{q}\sum_{t=0}^{T}[(y_t^{(r)} - Cu_{t|T}^{(r)})(y_t^{(r)} - Cu_{t|T}^{(r)})' + CP_{t|T}^{(r)}C')]$$

In the M-step, we optimize for the three terms separately. When an analytical solution is difficult to compute, we use gradient descent with back-tracking line search (2) (e.g., minimizing $g = \mathcal{L}_2$ and $g = \mathcal{L}_3$, where $x = \{A_t\}_{t=1}^{T}$ or $x = \{\sigma_i\}_{i=0}^{p}$.

---

**Data**: function $g$, initial value $x_0$, $\beta \in (0,1)$
**Result**: $x$ that minimizes $g(x)$
initialization: $x \leftarrow x_0$;
**while** *the difference of $x$ in two consecutive iterations is not small enough* **do**
    $\tau \leftarrow 1$;
    compute the gradient $\nabla g(x)$;
    **while** $g(x - \tau\nabla g(x)) > g(x) - \tau/2\|\nabla g(x)\|_2^2$ **do**
        $\tau \leftarrow \beta\tau$ ;
    **end**
    $x \leftarrow x - \tau\nabla g(x)$;
**end**

**Algorithm 1:** Backtracking with line-search

---

**Solving for dynamic connectivity parameters when state-variables are observed**

Given $\{u_t^{(r)}\}_{t=0,r=1}^{T,q}$, we solve for $Q_0$, $Q$ and $\{A_t\}_{t=1}^{T}$ by maximizing the log-likelihood plus the logarithm of the prior, which is equivalent to minimizing

$$-\log f(\{u_t^{(r)}\}_{t=0,r=1}^{T,q}) - \log f(\{A_t\}_{t=1}^{T})$$

$$\propto q\log\det(Q_0) + tr(Q_0^{-1}\sum_{r=1}^{q}u_0^{(r)}u_0^{(r)'})$$

$$+qT\log\det(Q) + tr(Q^{-1}\sum_{t=1}^{T}\sum_{r=1}^{q}(u_t^{(r)} - A_t u_{t-1}^{(r)})(u_t^{(r)} - A_t u_{t-1}^{(r)})')$$

$$+\lambda_0\sum_{t=1}^{T}\|A_t\|_F^2 + \lambda_1\sum_{t=2}^{T}\|A_t - A_{t-1}\|_F^2$$

where the optimization procedure is similar to that in the M-step. $Q_0$ has an analytical solution $Q_0 \leftarrow (1/q)\sum_{r=1}^{q}u_0^{(r)}(u_0^{(r)})'$, and $\{A_t\}_{t=1}^{T}$ and $Q$ can be updated in alternations. Given $\{A_t\}_{t=1}^{T}$, $Q$ has an analytical solution $Q \leftarrow 1/(qT)\sum_{t=1}^{T}\sum_{r=1}^{q}(u_t^{(r)} - A_t u_{t-1}^{(r)})(u_t^{(r)} - A_t u_{t-1}^{(r)})'$, and given $Q$, $\{A_t\}_{t=1}^{T}$ can be solved by gradient descent with backtracking line search, where the gradient is

$$2Q^{-1}(-\sum_{r=1}^{q}u_t^{(r)}(u_{t-1}^{(r)})' + A_t\sum_{r=1}^{q}u_{t-1}^{(r)}(u_{t-1}^{(r)})') + 2D_t$$

and

$$\boldsymbol{D}_t = \begin{cases} \lambda_1(2\boldsymbol{A}_t - \boldsymbol{A}_{t+1} - \boldsymbol{A}_{t-1}) + \lambda_0 \boldsymbol{A}_t & \text{for } t = 2, \cdots, T-1; \\ \lambda_1(\boldsymbol{A}_1 - \boldsymbol{A}_2) + \lambda_0 \boldsymbol{A}_1 & \text{for } t = 1; \\ \lambda_1(\boldsymbol{A}_T - \boldsymbol{A}_{T-1}) + \lambda_0 \boldsymbol{A}_T & \text{for } t = T; \end{cases}$$

**Computing the spatio-temporal covariance of the ROI mean activity and evaluating the marginal log-likelihood of sensor data**

According to the auto-regressive model, given $\{\boldsymbol{A}_t\}_{t=1}^T, \boldsymbol{Q}_0, \boldsymbol{Q}$, we have

$$\boldsymbol{u}_t = \boldsymbol{A}_t \boldsymbol{u}_{t-1} + \boldsymbol{\epsilon}_t = (\prod_{\tau=t}^1 \boldsymbol{A}_\tau)\boldsymbol{u}_0 + \boldsymbol{\epsilon}_t + \sum_{j=1}^{t-1}(\prod_{\tau=t}^{t-j+1} \boldsymbol{A}_\tau)\boldsymbol{\epsilon}_{t-j}$$

and the marginal covariance of $\boldsymbol{u}_t$ for $t = 1, \cdots, T$ is

$$\text{cov}(\boldsymbol{u}_t) = (\prod_{\tau=t}^1 \boldsymbol{A}_\tau)\boldsymbol{Q}_0(\prod_{\tau=t}^1 \boldsymbol{A}_\tau)' + \boldsymbol{Q} + \sum_{i=1}^{t-1}(\prod_{\tau=t}^{t-j+1} \boldsymbol{A}_\tau)\boldsymbol{Q}(\prod_{\tau=t}^{t-j+1} \boldsymbol{A}_\tau)'$$

where the in the product $\prod_{\tau=t}^{t-j+1}$, $\tau$ decreases from $t$ to $t-j+1$. To compute the marginal covariance, it is convenient to first compute matrices $\tilde{\boldsymbol{A}}_{j,k} = \prod_{\tau=k}^j \boldsymbol{A}_\tau, j \leq k$, and then we have

$$\text{cov}(\boldsymbol{u}_t) = \tilde{\boldsymbol{A}}_{1,t}\boldsymbol{Q}_0\tilde{\boldsymbol{A}}'_{1,t} + \boldsymbol{Q} + \sum_{j=1}^{t-1}\tilde{\boldsymbol{A}}_{(t-j+1),t}\boldsymbol{Q}\tilde{\boldsymbol{A}}'_{(t-j+1),t}.$$

Let $\boldsymbol{U}_{p \times (T+1)} = [\boldsymbol{u}_0, \boldsymbol{u}_1, \cdots, \boldsymbol{u}_T]$ (where $\boldsymbol{u}_t$ is of size $p \times 1$); let $\text{vec}(\boldsymbol{U})$ be the concatenation of the columns of $\boldsymbol{U}$. Let the $p(T+1) \times p(T+1)$ matrix $\boldsymbol{\Sigma}$ denote the covariance matrix of $(\text{vec}(\boldsymbol{U}))$. This covariance can be computed as

$$\boldsymbol{\Sigma}[tp+1 : (t+1)p, (t+h)p+1 : (t+h+1)p]$$
$$= \text{cov}(\boldsymbol{u}_t, \boldsymbol{u}_{t+h})$$
$$= \text{cov}(\boldsymbol{u}_t, \boldsymbol{A}_{t+h}\boldsymbol{A}_{t+h-1} \cdots \boldsymbol{A}_{t+1}\boldsymbol{u}_t)$$
$$= \text{cov}(\boldsymbol{u}_t)(\prod_{\tau=t+h}^{t+1} \boldsymbol{A}_\tau)'$$
$$= \text{cov}(\boldsymbol{u}_t)\tilde{\boldsymbol{A}}'_{t+1,t+h}$$

where by $\boldsymbol{\Sigma}[tp+1 : (t+1)p, (t+h)p+1 : (t+h+1)p]$, we mean the sub-matrix composed of the consecutive rows from $tp+1$ to $(t+1)p$ and the consecutive columns from $(t+h)p+1$ to $(t+h+1)p$ (indices are 1-based). If needed, we can also compute the marginal correlation between the mean of $i_1$th ROI at time $t_1$ and the mean of $i_2$th ROI at time $t_2$:

$$\text{correlation}(\boldsymbol{u}_{t_1}[i_1], \boldsymbol{u}_{t_2}[i_2]) = \frac{\boldsymbol{\Sigma}[t_1 p + i_1, t_2 p + i_2]}{\sqrt{\boldsymbol{\Sigma}[t_1 p + i_1, t_1 p + i_1]\boldsymbol{\Sigma}[t_2 p + i_2, t_2 p + i_2])}} \tag{9}$$

Next, we discuss how to evaluate the objective function that contains the marginal likelihood of the observed multi-trial sensor data and a prior term for the parameters (i.e., $\log f(\{\boldsymbol{y}_t^{(r)}\}_{t=0,r=1}^{T,q}) + \log f(\boldsymbol{\theta})$ ). Since $\log f(\boldsymbol{\theta})$ is easy to compute, we mainly focus on evaluating $-\log f(\{\boldsymbol{y}_t^{(r)}\}_{t=0,r=1}^{T,q})$. Let the $n \times (T+1)$ matrix $\boldsymbol{Y} = (\boldsymbol{y}_0, \boldsymbol{y}_1, \cdots, \boldsymbol{y}_T)$ denote the sensor time series in an arbitrary trial. Let $\text{vec}(\boldsymbol{Y})$ be a vector obtained by concatenating the columns in $\boldsymbol{Y}$.

Let $\tilde{\boldsymbol{C}}$ be an $nT \times pT$ block-diagonal matrix, where $\tilde{\boldsymbol{C}}[tn+1 : (t+1)n, tp+1 : (t+1)p] = \boldsymbol{C}$, for $t = 0, 1, \cdots, T; \tilde{\boldsymbol{C}}[tn+1 : (t+1)n, tp+1 : (t+1)p]$ denotes the intersection of the sub-rows (rows $tn+1$ to $(t+1)n$) and the sub-columns (columns $tp+1$ to $(t+1)p$). Similarly, let $\tilde{\boldsymbol{R}}$ be an $nT \times nT$ block diagonal matrix, where $\tilde{\boldsymbol{R}}[tn+1 : (t+1)n, tn+1 : (t+1)n] = \boldsymbol{R}$, for $t = 0, 1, \cdots, T$. Then we have

$$\text{cov}(\text{vec}(\boldsymbol{Y})) = \tilde{\boldsymbol{C}}\boldsymbol{\Sigma}\tilde{\boldsymbol{C}}' + \tilde{\boldsymbol{R}}$$

and

$$-\log f(\{\boldsymbol{y}_t^{(r)}\}_{t=0,r=1}^{T,q}) \propto q \log \det(\mathrm{cov}(\mathrm{vec}(\boldsymbol{Y}))) + \sum_{r=1}^{q} tr(\mathrm{cov}(\mathrm{vec}(\boldsymbol{Y}))^{-1}\mathrm{vec}(\boldsymbol{Y}^{(r)})\mathrm{vec}(\boldsymbol{Y}^{(r)})')$$

$+$ some constant irrelevant to the parameters

We use the following equations to facilitate the computation

$$\mathrm{cov}(\mathrm{vec}(\boldsymbol{Y}))^{-1} = \tilde{\boldsymbol{R}}^{-1} - \tilde{\boldsymbol{R}}^{-1}\tilde{\boldsymbol{C}}(\boldsymbol{\Sigma}^{-1} + \tilde{\boldsymbol{C}}'\tilde{\boldsymbol{R}}^{-1}\tilde{\boldsymbol{C}})^{-1}\tilde{\boldsymbol{C}}'\tilde{\boldsymbol{R}}^{-1}$$

$$\log \det(\mathrm{cov}(\mathrm{vec}(\boldsymbol{Y}))) = \log \det(\boldsymbol{\Sigma}^{-1} + \tilde{\boldsymbol{C}}'\tilde{\boldsymbol{R}}^{-1}\tilde{\boldsymbol{C}}) + \log \det(\boldsymbol{\Sigma}) + \log \det(\tilde{\boldsymbol{R}})$$

**Estimating the source space activity**

Estimating the source space activity $\boldsymbol{J}_t$ is not our primary goal in this model. However, we can still estimate $\boldsymbol{J}_t$ once the ROI means $\boldsymbol{u}_t$ is estimated in the end of the E-M algorithm. Given the hierarchical model below

$$\boldsymbol{J}_t|\boldsymbol{u}_t \sim \mathcal{N}(\boldsymbol{L}\boldsymbol{u}_t, \boldsymbol{Q}_J)$$
$$\boldsymbol{y}_t|\boldsymbol{J}_t \sim \mathcal{N}(\boldsymbol{G}\boldsymbol{J}_t, \boldsymbol{Q}_e)$$

we have

$$\boldsymbol{J}_t|(\boldsymbol{u}_t, \boldsymbol{y}_t) \sim \mathcal{N}\left((\boldsymbol{G}'\boldsymbol{Q}_e^{-1}\boldsymbol{G} + \boldsymbol{Q}_J^{-1})^{-1}(\boldsymbol{G}'\boldsymbol{Q}_e^{-1}\boldsymbol{y}_t + \boldsymbol{Q}_J^{-1}\boldsymbol{L}\boldsymbol{u}_t), \quad (\boldsymbol{G}'\boldsymbol{Q}_e^{-1}\boldsymbol{G} + \boldsymbol{Q}_J^{-1})^{-1}\right) \tag{10}$$

Given an estimate of $\boldsymbol{u}_t$, which we denote by $\hat{\boldsymbol{u}}_t$, we can estimate $\boldsymbol{J}_t$ by the mean term in Equation 10,

$$\hat{\boldsymbol{J}}_t = (\boldsymbol{G}'\boldsymbol{Q}_e^{-1}\boldsymbol{G} + \hat{\boldsymbol{Q}}_J^{-1})^{-1}(\boldsymbol{G}'\boldsymbol{Q}_e^{-1}\boldsymbol{y}_t + \hat{\boldsymbol{Q}}_J^{-1}\boldsymbol{L}\hat{\boldsymbol{u}}_t)$$

where $\hat{\boldsymbol{Q}}_J$ is based on the estimations of $\sigma_i^2$s. Note that to make the notation succinct, we have omitted the trial index $^{(r)}$. But such an estimate step can easily be done for each time point $t$ and each trial.

**Additional details about the simulations**

In the simulations, the bilateral pericalcarine sulci and the parahippocampal gyri were automatically labeled by the Freesurfer software (3) according to the anatomy. With $p = 2$ ROIs, $T = 20$ time points and $q = 200$ trials, we had a relatively large number of observations. Therefore, in the prior of $\{\boldsymbol{A}_t\}_{t=1}^T$, we preselected $\lambda_0 = 0$ and $\lambda_1 = 0.1$, expecting that these values only added minimal regularization. In practice, these values can be selected as the ones that maximize the cross-validated log-likelihood of the sensor data. When using the posterior estimates of $\boldsymbol{u}_t$ to predict $\boldsymbol{y}_t$ and quantify how much variance of the sensor data was explained by the state-variables, we observed about $5.95\%$ in our simulations when $a = 2$ and $10.84\%$ when $a = 5$.

It is worth noting that the in the two-step MNE method, although the estimated $\boldsymbol{u}_t$ could be temporally correlated with the true $\boldsymbol{u}_t$ in the simulations, the scale was much smaller. For example, Figure 1a shows the true $\boldsymbol{u}_t$ (blue) in one trial, one ROI (ROI 2), and one simulation ($\alpha = 5$), as well as the estimated $\boldsymbol{u}_t$ in the two-step MNE method (red), and the posterior estimate of $\boldsymbol{u}_t$ by the state-space model (green). The one from the two-step MNE method was close to zero. This could be due to the shrinkage effect of the $L_2$ penalty with a relatively large penalization parameter $\lambda$. However, using a smaller $\lambda$ resulted in extremely noisy estimations of $\boldsymbol{u}_t$, which were not correlated with the true $\boldsymbol{u}_t$.

Although the auto-regressive (AR) coefficients ($\boldsymbol{A}_t$s) should not be very sensitive to the scales of $\boldsymbol{u}_t$, when the scaling is different for each ROI, the off-diagonal entries might be affected. In contrast, the diagonal entries, which describe the dependence of the ROI activity on the one-step-back history of itself, are unlikely to be affected by the scales. To give the two-step MNE method a chance to correct for scaling for the off-diagonal entries, we also computed a different measurement of the relative error. Let $\boldsymbol{A}_t$ denote the true AR coefficients. For each pair of ROIs, $(i, j)$, we multiply the estimates ($\hat{\boldsymbol{A}}_t$) with a scalar $\alpha$, such that $\sum_{t=1}^T (\boldsymbol{A}_t[i,j] - \alpha\hat{\boldsymbol{A}}_t[i,j])^2$ was minimized, and then we defined the relative error as

$$\sqrt{\frac{\sum_{t=1}^T (\boldsymbol{A}_t[i,j] - \alpha\hat{\boldsymbol{A}}_t[i,j])^2}{\sum_{t=1}^T (\boldsymbol{A}_t[i,j])^2}}$$

We took the average of the relative error after scaling for the off-diagonal $(i, j)$ pairs (i.e. $i \neq j$) (Figure 1b); in the results, we still observed that the state-space model yielded smaller errors than the two-step MNE method.

(a)

(b)

Figure 1: (a), Visualization of $\boldsymbol{u}_t$ in ROI 2 in one trial from a simulation. (The legends: truth (blue), true values; ss (green), estimates by the state-space model; mne (red), estimates by the two-step MNE method.) (b) Relative error of AR coefficients after scaling, averaged within off-diagonal entries by the state-space models and the two-step MNE method, as well as the paired difference. The error bars show standard errors across individual simulations.

**Additional details about the empirical MEG data**

Data in the MEG experiment was collected using a 306-channel whole-head MEG system (Elekta Neuromag, Helsinki, Finland) at the Brain Mapping Center at the University of Pittsburgh. During the experiment, a human participant processed naturalistic photographs of scenes. In each trial, while the participant fixated their eyes on a "+" symbol in the center of a gray screen, one image centered in the screen appeared and lasted for 200 ms. Afterwards, the screen switched back to the "+" symbol on the gray background. The participant was instructed to press a button if the image was identical to the previous one. In the entire session, 362 images of scenes were used and each image was repeated 5 times (excluding the consecutive repetitions) in a pseudo-random order. The participant gave written informed consent and was financially compensated. All procedures followed the principles in the Declaration of Helsinki and were approved by the Institutional Review Boards of Carnegie Mellon University and the University of Pittsburgh.

The MEG recordings were acquired at 1 kHz, high-pass filtered at 0.1 Hz and low-pass filtered at 330 Hz. In the preprocessing, the raw data was filtered with a 1-110 Hz bandpass filter, and then with a notch filter at 60 Hz to reduce the power-line interference; independent component analysis (ICA) was used to decompose the MEG data into multiple components, and the components that were highly correlated with eye blinks and heartbeats were removed. After the preprocessing, the MEG data was cropped from -140 ms to 960 ms for each trial, where 0 is the stimulus onset. A signal space projection (SSP) was applied to the data; the SSP constructed a low-dimensional linear subspace characterizing the empty room noise (via principal component analysis), and removed from the experimental MEG recordings the projection onto this subspace, so that neural signals orthogonal to the principal components of empty room noise remained. For each sensor, a baseline temporal mean within -140 ms to -40 ms was subtracted for all time points, separately for each trial. The data was further down-sampled at 100 Hz (i.e., one time point for every 10 ms). For each image, data across 5 repetitions were averaged, and further cropped from $-100$ to 700 ms. Finally, the sample mean of the $n \times (T + 1)$ sensor data across the $q = 362$ observations were subtracted off. The preprocessing, the source-space definition, the forward modeling, and the minimum-norm estimates were done in the MNE (MNE-python) software (4; 5).