[Reviews · NeurIPS 2016]

Reviewer 1

Summary

This paper addresses the problem of localization active brain regions from MEG and EEG but at the same time estimating the "connectivity" between brain regions (yet from a predefined list of regions which makes the problem not so ill-posed). While most inverse methods assume independence between brain regions, the state space model proposed aims to do everything in one step, which is a particularly appealing idea. The model makes use of time-varying auto-regressive (TVAR) models to capture non-stationarities which is also particularly relevant for such data. To further regularize the estimation, the AR coefficients (A_t)_t are regularized with a Frobenius norm and the Frobenius norm of the difference of A_t between neighboring time points. Inference is done with EM. The method is compared to the 2 steps approach using L2 regularization aka MNE on a simulation dataset. The method is then evaluated on MEG data on a protocol involving visual scene presentation.

Qualitative Assessment

Comments: - a time varying AR1 model is used and data were downsampled at 200Hz. As the order of the filter interacts with the sampling rate, did this choice affect the conclusions obtained from the model. - results obtained within the selected labels are commented. What happens and what is estimated on the remaining part of the brain is unclear. - it is unclear how much the proposed model fits the data and we don't know how much variance is actually explained by the estimated sources / network. - Discussion about the limitations of the method is appreciated. Yet the sample complexity of the method is not really discussed. Given the number of paramters to estimate how many pre-defined regions and samples are reasonable to infer? Missing ref: Z. Zhang and B. D. Rao, “Sparse signal recovery with temporally correlated source vectors using sparse bayesian learning,” IEEE J. Sel. Topics Signal Process., vol. 5, no. 5, pp. 912–926, Sept. 2011. Misc: - l200 : the amplitude of equivalent current dipoles in MEG/EEG inverse modeling is nanoAmpere Meter (nAm) as it's the unit of electric dipole moment. The term Ampere comes the physicist André-Marie Ampère. - in the bibliography make sure acronyms such as EEG, MEG, EM, etc. are rendered in capital letters. - l142 : and for -> for

Confidence in this Review

3-Expert (read the paper in detail, know the area, quite certain of my opinion)


Reviewer 2

Summary

This paper presents a probabilistic model-based method to analyze dynamic connectivity between regions of interest (ROIs) from MEG/EEG, with the source localization step embedded. The mean activities of ROIs are assumed to follow a multivariate autoregressive process, under which source activities in each ROI are distributed as isotropic Gaussian. The autoregressive source model is combined with electromagnetic forward (measurement) model so that the entire model is a specific type of state-space model. The EM algorithm is used for model fitting. The method is first validated in a simulation, in which the state-space method leads to better estimation accuracy as compared to conventional two-step method based on minimum-norm source estimates. A preliminary analysis result about real MEG data on scene processing is also presented.

Qualitative Assessment

This is a well-written paper about an interesting integrative approach to dynamic connectivity analysis in MEG/EEG. The method is well-motivated, clearly presented and developed in a mathematically sound way. Concerning related work, the following recent paper (not cited) also presented a similar model: M. Fukushima et al., MEG source reconstruction based on identification of directed source interactions on whole-brain anatomical networks, NeuroImage, 105:408-427, 2015. This also uses a state-space model for MEG source localization, in which spatially long-range connections are considered. Some comments on this previous work will be necessary to clarify the novelty of the present study (e.g. ROI vs spatial smoothing, time-varying or not, etc.). In Section 3, the lambda, lambda_0 and lambda_1 values seem to be fixed at constants. How were these values selected? How robust are those results to the choice of these values? Some comments on any practical methods to determine these values will also be needed. In the MEG result, what does the small cluster at the top right of each \Delta matrix (after 600ms) means? This is also remarkable in the result and should also be discussed. I also wonder how the nonstationarity assumption (time-varying connections) actually affected the result. It would be interesting to compare the result with stationary model. The authors mention about both phase-locking values and Granger causality (line 53) as the examples of connectivity measure. Does the proposed state-space model actually apply to connectivity measures other than Granger causality? Introduction should explain more clearly about what kinds of connectivity analysis the proposed method can perform.

Confidence in this Review

2-Confident (read it all; understood it all reasonably well)


Reviewer 3

Summary

Great paper! In this contribution, the authors propose a brand new approach for assessing dynamic functional connectivity in the source space from MEG recordings. Instead of considering a two step approach that first proceeds to source localization and then to connectivity matrix estimation in the source space, the authors develop a state space approach with time varying AR parameters and ROI activity means as latent variables to jointly perform these two steps. The method is definitely well evaluated on intensive and well studied simulations and interesting validation is proposed on a single individual for analyzing how visual information flows in the brain in response to the presentation of image scenes.

Qualitative Assessment

Very nice contribution as compared to classical incremental research in the MEG functional connectivity field. A question: How can you guarantee the stability of the AR filter at each time point? A few minor remarks: - specify from the beginning that the ROIs are defined in the source space - Perhaps try to refer the seminal paper in the 90s about inference of time-varying AR models with EM algorithm! - underspecified -> underdetermined (hence ill-posed problem) - Comment the results in the top right corner of the matrix appearing in Fig. 3c: why did you find top-down feedback connection between EVP and PPA using the two-step approach? - list of typos (p.2 line 78: p. 4 line 142; p.5 line 194 + preserve capital letters in acronyms in the bibliography section. This remark also holds for your appendix).

Confidence in this Review

3-Expert (read the paper in detail, know the area, quite certain of my opinion)


Reviewer 4

Summary

The authors propose a state-space model to estimate the cross-region dynamic connectivity in MEG and EEG data. Instead of common methods which take two independent steps, the authors try to solve this problem within one model by taking the regions of interest, the sources and the sensors together, and then solve the model by EM algorithm. Quantitative results on synthetic dataset and visualization on real MEG dataset are shown to compare the proposed state-space model with two step minimum-norm estimate baseline.

Qualitative Assessment

The idea of jointly solving the cross-region connectivity within one model is quite interesting and reasonable. The authors provided clear descriptions of the model and detailed discussions on both the pros and cons of proposed method. The results are also promising, and it is interesting to see some new finding in the experiments from the new method. I only have some comments. The authors assume that all sources within one region of interests share the same mean activity value, which might be true or not in real application. One thing to mention is how the number and locations of ROIs are determined, and this may support the above assumptions. The authors provided quantitative simulation results and visualizations on both simulation and real MEG data. The proposed model achieves smaller estimation errors on simulation, and the visualization results also demonstrate the better performance of the one-step model. My major concern on this part is about the computational cost. The two step baseline is similar to doing one iteration of EM, and thus the efficiency comparison is helpful to evaluate whether the proposed method is more applicable. I’m glad to see more empirical evaluations on other EEG datasets.

Confidence in this Review

2-Confident (read it all; understood it all reasonably well)


Reviewer 5

Summary

Authors propose a state-space model for calculating and estimating dynamic functional connectivity in MEG and EEG. The state variable is a mean time course of measured activities within each ROI. State variables are converted into source space, and then into sensor space. Each conversion is performed by source and sensor model, respectively. Model parameters are optimized by the expectation-maximization (EM) algorithm. They demonstrated the effectiveness of their proposed model on both simulated and measured MEG data.

Qualitative Assessment

Analytical framework using state-space model proposed by authors is novel and very interesting. Their model is very simple but has great impact on the field of functional connectivity analysis. However, it is regrettable that their analysis were performed in terms of comparison with conventional two-step method. I think that analysis and discussion on new direction or aspects indicated by the results obtained from their model are more desirable and interesting. For example, how do the time-varying auto-regressive coefficients (At) vary on different dataset? Is it able to represent the dynamic behavior of brain network shifting between known functional networks (DMN, CEN, SN, ...)? I hope it will be performed as future work.

Confidence in this Review

2-Confident (read it all; understood it all reasonably well)


Reviewer 6

Summary

The paper elegantly presents an approach to source localization that is applicable to both MEG and EEG data (and other modalities), highly sensitive to temporal variations and allows direct extraction of statistical dependance and time lags, from a set of mutually correlated signals (linearly generated from sources). One of the main attributes advocated in this work is that separation into statistically separable sources need not be done independently of causality, or that of time-lag analysis of sources. Rather, the authors present a single model that decomposes the data into sources and models their time dependence, based on expectation-maximization approach. An application to MEG analysis is presented.

Qualitative Assessment

The Introduction is generally very good (with minor exceptions described below). Comparison to other models is required. Only one alternative approach is compared to the suggested method and another one-step model (DCM) is not lawfully described. I suggest the authors discuss other applications beside EEG/MEG as many of the alternative approaches were shown to be useful to many modalities. Please introduce consistent spacing before citations (in many cases the space doe not exist at all). Specific comments: 1. 46. Perhaps the authors can also point to the ill-poshness of the problem due to SNR and destructive local activities: even if we had the number of observation roughly equal to the number of sources still unravelling source activity from scalp dynamics would have been a major challenge. Please elaborate if this is not the case. 2. 54. Please expand “based on different assumptions.” or if those assumptions are already mentioned before, say it. 3. 60. Perhaps the authors can add a reference to the claim? 4. 61. Is this claim addressed later? If yes, say so, if not, expand this claim, or give proper reference, or omit this claim altogether. 5. 77. describe the model “in detail”. 6. 69-71. The text seems to undermine Ref 1. Please explain why DCM is irrelevant to the claim. If it’s relevant please expand on this issue. 7. 85. “includes”. 8. 94. In what neighborhood, of the sensor? Not clear: please revise the entire sentence, including the improper use of “according to the physics” (92), as the discussed notions combine physical (and biophysical) attributes with anatomical ones. 9. 100 “Gaussian distribution”. 10. 108. “The state variable u_t…” In general I suggest sentences won’t start with symbols. 11. The definition of A_0 should be fixed. It doesn’t make sense to say $l$ and $\forall l$ — Please make it more formal. 12. 122. “Equations…” 13. 123. The way the sentence is written should be “is a 0/1…”. Later the authors can omit “in the lth row and ith column”. 14. 123. The way Q_J is described is not a proper definition (not clear what is the exact value of Q_J for the case corresponding to equation 4). In fact the “only if” could be a wrong assertion, if I understand correctly what is meant by the authors. 15. I don’t see how the authors “integrated J_t out.” Isn’t it just a substitution? 16. 132-134 is clear but informal. Please improve the text by making the definitions more formal/correct. 17. 134. Do you need the square when designating the Forbenuis norm? Isn’t the norm given by the notation || ||_F ? 18. 142. “and” is repeating twice. 19. 190. Typo: “pericalcarine suci.” 20. 194: Typo: “entries was.” 21. 198-199. It’s not so clear what is meant by “were different.” 22. 200. Do you mean “nanoamper”? 23. 215. “which underlines recognition of what we see” is not entirely clear and also a too bold description. 24. 244-247. The statistical analysis is not presented in a clear manner. It is also not clear how the p-values are derived given that the T^2 observations are statistically dependent. Please make this part clearer and make sure the assumptions of the statistical test are met. 25. 289. The reference to “cognitive processes” as an exceptional case is not clear, whereas the authors suggest a simple dynamical behavior for other neural processes. The DCM model should be extended also here (and earlier in the paper). 26. 290. The reasoning for not comparing against DCM is weak and should be improved (“Part of of the reason…” — the comparison is required). The authors should make efforts to directly compare their results to DCM. Ref 1. David et al. “Dynamic causal modeling of evoked responses in EEG and MEG”

Confidence in this Review

2-Confident (read it all; understood it all reasonably well)